# Small Diameter Cell-Free Tissue-Engineered Vascular Grafts: Biomaterials and Manufacture Techniques to Reach Suitable Mechanical Properties

**DOI:** 10.3390/polym14173440

**Published:** 2022-08-23

**Authors:** María A. Rodríguez-Soto, Camilo A. Polanía-Sandoval, Andrés M. Aragón-Rivera, Daniel Buitrago, María Ayala-Velásquez, Alejandro Velandia-Sánchez, Gabriela Peralta Peluffo, Juan C. Cruz, Carolina Muñoz Camargo, Jaime Camacho-Mackenzie, Juan Guillermo Barrera-Carvajal, Juan Carlos Briceño

**Affiliations:** 1Department of Biomedical Engineering, Universidad de los Andes, Bogotá 111711, Colombia; 2Vascular and Endovascular Surgery Research Group, Fundación Cardio Infantil—Instituto de Cardiología, Bogotá 111711, Colombia; 3School of Medicine and Health Sciences, Universidad del Rosario, Bogotá 111711, Colombia; 4Department of Research, Fundación Cardio Infantil Instituto de Cardiología, Bogotá 111711, Colombia

**Keywords:** tissue engineered vascular grafts, biomaterials, biodegradable, biomechanical stimulation, mechanical properties

## Abstract

Vascular grafts (VGs) are medical devices intended to replace the function of a blood vessel. Available VGs in the market present low patency rates for small diameter applications setting the VG failure. This event arises from the inadequate response of the cells interacting with the biomaterial in the context of operative conditions generating chronic inflammation and a lack of regenerative signals where stenosis or aneurysms can occur. Tissue Engineered Vascular grafts (TEVGs) aim to induce the regeneration of the native vessel to overcome these limitations. Besides the biochemical stimuli, the biomaterial and the particular micro and macrostructure of the graft will determine the specific behavior under pulsatile pressure. The TEVG must support blood flow withstanding the exerted pressure, allowing the proper compliance required for the biomechanical stimulation needed for regeneration. Although the international standards outline the specific requirements to evaluate vascular grafts, the challenge remains in choosing the proper biomaterial and manufacturing TEVGs with good quality features to perform satisfactorily. In this review, we aim to recognize the best strategies to reach suitable mechanical properties in cell-free TEVGs according to the reported success of different approaches in clinical trials and pre-clinical trials.

## 1. Introduction

Cardiovascular diseases (CVDs) are a group of conditions affecting the heart and blood vessels. According to the World Health Organization, CVDs are currently the leading cause of death worldwide, representing 32% of all global deaths [1]. Accordingly, almost 18 million people died from a CVD in 2019. Among CVDs, vascular diseases compromising the blood vessel structure take hold of more than 8.5 million people worldwide. For instance, coronary artery disease (CAD) is the most common type of CVD, affecting 6.7% worldwide. CAD alters the mechanical properties of the coronary artery supplying the heart muscle, resulting in a decrease in the blood flow [2]. Moreover, other conditions also affect the mechanical properties of the blood vessels, such as the Peripheral Artery Disease (PAD), which compromises the blood flow in the limbs, and aortic aneurysms, identified by a balloon-like bulge in the aorta [3].

Currently, surgical approaches for vascular reconstruction focus on using autologous or synthetic vascular grafts (VGs) to recover the blood flow. Autologous grafts such as the great saphenous vein (GSV) and Internal Mamary Artery (IMA) can maintain long-term patency due to their regenerative properties (86% up to five years). However, GSV has limited availability and represents additional risks due to possible post-surgical complications with morbidity rates up to 50% [4]. Furthermore, synthetic VGs in the market are made from Polytetrafluoroethylene (PTFE—known as Teflon^®^) and Polyethylene terephthalate (PET—known as Dacron^®^). Although their performance is suitable in straight blood vessels and large diameters, they are inefficient in replacing small diameter vessels (>6 mm diameter) and under non-unidirectional or non-fully developed flows, as seen in their decreased patency rates below 32% after two years [5]. These low patency rates have been correlated to a foreign body response promoting thrombogenesis and stenosis, compromising the blood flow. With 3 out of 100,000 adults requiring peripheral vascular segment repairs through small diameter VGs [6], new approaches to overcome the current challenges include the use of bio-based VGs, such as allogeneic cryopreserved blood vessels (Cryograft^®^—CryoLife) [7], xenografts from bovine blood vessels (Artegraft^®^—LeMaitre) [8], bovine pericardium [9], or xenografts from sheep extracellular matrix (ECM) (Omniflow^®^ II—LeMaitre) [10]. Nevertheless, they still do not show a clear advantage compared to commercial synthetic non-biodegradable VGs [11].

These bio-based VGs are part of Tissue-engineered vascular grafts (TEVGs); these scaffolds are intended to guide the tissue regeneration of the vascular wall and are considered as a strategy to get closer to the biological response of native blood vessels. Under physiological conditions, native arteries have a complex, multilayered structure with different ECM compositions and microstructures to contribute to the required compliance precisely responding to pulsatile pressure. Components in each layer will determine the total vessel elastic, viscous, and inertial properties that provide the critical biomechanical signals required to regulate cell adhesion, growth, and differentiation [12].

Although there is promising research on TEVGs, several barriers are still to be overcome. In this sense, the rational design of TEVGs must consider the self-defeating cellular response to biomaterial surfaces and structures under the hemodynamic operative conditions leading to graft patency loss. Accordingly, the inflammatory process after implantation of a TEVG is required for the vascular wall regeneration and depends on the ECM composition and mechanical properties regarding the presence of bioactive molecules and their micro/macrostructure [13]. For instance, one of the most reported causes for patency loss in TEVGs arises due to differences between the mechanical properties of the TEVGs and the native vessels affecting the flow pattern and the biomechanical stimulation. Furthermore, the inability of the cells to infuse into the scaffold from the perivascular tissues can also promote failure [14,15].

The failure of TEVGs depends on the alteration of the complex interactions between the biomaterials, cells, and hemodynamic conditions in the blood flow. Namely, the most reported failure of TEVGs is related to the development of intimal hyperplasia, in which smooth muscle cells over-proliferate, and also atherogenesis, in which foreign body responses promote graft calcification [16]. However, other failure causes include aneurysms, thrombogenesis, and bacterial infection, affecting the flow pattern and the biomechanical stimulation. Consequently, for a TEVG to provide continuity between the native tissue and the required biomechanical signals to induce regeneration, the rational design must consider the suitable mechanical properties of biodegradable structures and biomaterials similar to the native arteries in which the TEVG will be anastomosed, and to the current gold standards (i.e., GSV and IMA) [17].

Herein, we aim to recognize the best strategies to reach suitable mechanical properties in TEVGs according to the success of different approaches in clinical and pre-clinical trials. Furthermore, we aim to identify the latest trends in cell-free TEVGs development regarding the manufacturing methods and biomaterials in the context of ideal mechanical properties.

## 2. Database Information Extraction and Data Analysis

An integrative review of the literature on tissue-engineered vascular grafts was conducted and reported according to the elements described in the PRISMA guidelines. To this end, an electronic search was performed in the following databases: MEDLINE, SCOPUS, Web of Knowledge, and ClinicalTrials.gov. The indexing terms for the search strategy were in MeSH terminology. These terms included “Tissue engineering”, “Vascular graft”, “Biodegradable” and “Regenerative” from the last five years (2017–2022). A total of 4592 records were identified, with 2476 removed before the screening, as they were considered duplicated records or ineligible by automation tools.

For the selection process, an initial title review was performed; if the title indicated that the study might be relevant, abstracts were reviewed, or they were otherwise excluded. Finally, identified eligible studies were read on a full-text basis. Inclusion and exclusion criteria were applied to the list of articles obtained; those that did not meet these criteria were eliminated, and the documents corresponding to the remaining articles were reviewed. Thereof, five registries in Clinicaltrials.gov were included for TEVGs, 43 records were included for TEVGs on pre-clinical models (Appendix A), and 46 studies found in databases were included for TEVGs on in vitro testing (Appendix A)

The inclusion criteria for the screening and choice were covered by TEVGs on clinical trials registered on clinicaltrials.gov, with pre-clinical testing on in vivo models and data related to in vitro studies found in different databases, reporting at least two mechanical properties according to the requirements of the ISO 7198:2016 and from the last five years; otherwise, manuscripts were excluded. Additional exclusion criteria considered TEVGs with pre-seeded cells, articles in languages different from English or Spanish, or if the retrieved information was incomplete, did not correspond to the scope of the review, or was unavailable for retrieval for any reason. A total of 2160 records were screened, 1745 were removed, and classification was performed considering three categories: TEVGs on clinical trials, TEVGs on pre-clinical models, and TEVGs with in vitro data.

Subsequently, data collection and extraction were performed regarding the mechanical properties reported in the TEVGs analyzed from the clinical trials, pre-clinical models, or in vitro models and the relationship with patency and regeneration potential. Reviewers had independent access and the data were fed as the literature review process was carried out. The data were classified according to the study type and organized according to title, authors, DOI or CT registry number, and year of publication, including manufacturing techniques, materials, and any functionalization used for developing the TEVG. Detailed information about the study, animal model, and physicochemical and biological characterization was incorporated, considering the regenerative potential through the endothelialization success and inflammatory response.

As the manuscripts reporting mechanical properties were included, we compared available biodegradable TEVGs to native blood vessels (GSV and IMA) as the least defining suitable mechanical properties to be fulfilled to the TEVGs. Then, all summarized data were analyzed using descriptive analysis, reporting central tendency measures for quantitative variables and frequencies for qualitative variables. Finally, all the information was tabulated. This article’s relevant data are in tables or graphics presented in this review.

## 3. Physiology of Blood Vessels

The vascular system is comprised of blood vessels that can be classified into arteries, capillaries, and veins, with differences according to their size and physiological function. Blood rich in oxygen and nutrients is transported from the heart to different tissues in the body by a network of capillaries, branching from the arteries. The capillaries then transport the blood containing carbon dioxide and other metabolic wastes from the tissues to the venules and veins to remove and re-oxygenate the blood in the lungs.

Besides their physiological differences, blood vessels have a complex vascular wall in terms of composition and structure, which gives them a wide variety of mechanical properties according to the different hemodynamic conditions to which the blood vessel can be exposed. However, these properties are potentially affected by pathologies that change the vascular wall, such as atherosclerosis or aneurysms, representing the world’s leading causes of death [12]. Currently, Coronary Artery Disease (CAD) and Peripheral Arterial Disease (PAD) represent the leading cause of mortality worldwide, with an estimated annual incidence increase of 23.3 million by 2030 [18]. Angioplasty, stent implantation, and surgical bypass grafting are the current treatment options. For the latter, autologous veins, or arteries such as the GSV or IMA, represent the gold standard for this and are preferred due to their long-term patency up to 80% at five years [18].

### 3.1. Micro and Macrostructure of Blood Vessels 

At the macrostructural level, the vascular wall of blood vessels presents a multilayered structure composed of 70% water and 30% collagen, elastin, proteoglycans, and vascular cells [12]. The vascular wall is distinguishable in three layers: tunica intima, media, and adventitia (Figure 1). The innermost layer–the tunica intima–is constituted by endothelial cells (ECs) as a monolayer on the blood vessel’s lumen and maintains homeostasis through a carefully regulated interaction with the other types of cells. Following this layer, and separated by an inner membrane composed of elastin, there is the tunica media composed mainly of smooth muscle cells (SMCs), elastin, and collagen fibers. Finally, the tunica adventitia is a layer of fibroblasts and other components of the extracellular matrix (EMC)–mainly collagen fibers [19].

Nevertheless, the wall thickness and its components may vary according to the type of blood vessel. Due to the aim of this study, we will focus on small arteries with diameters ranging between 1–6 mm and a wall thickness between 125–800 µm [12]. Figure 2 illustrates the composition of this kind of small-sized blood vessels.

### 3.2. Mechanical Properties of Blood Vessels Dsed as Gold Standards on Vascular Grafts

The mechanical analyses of blood vessels are of utmost importance to understand their properties under different hemodynamic conditions and to guide the rational design and production of TEVGs capable of withstanding physiological stresses and pressures. The blood vessel wall presents a complex structure and composition, allowing responses to different strengths and pressures caused by blood flow. Due to the anisotropic behavior of blood vessels, they support a significant load in the circumferential direction due to the collagen components in this direction, while allowing compliance given the elastin fibers. Therefore, at high pressures in the blood vessels, the elastin fibers will allow expansion, and collagen fibers will be stiffer to permit the change in diameter but prevent damage or rupture when the pressure increases. Moreover, the compliance of the blood vessels is a property that allows us to measure the storage of blood that the artery can support and release to the vascular network in order to reach regions of lower pressure in a pulsatile flow through its stretching due to the elastin fibers [12]. 

Currently, for the substitution of small diameter vessels (<6 mm), autografts from GSV and IMA are used due to their excellent compliance and compatibility with the native vessels, remaining as the gold standard [20]. For this study, GSV will be considered as a standard due to its great usefulness as a graft, being a blood vessel that presents a significant length and is easily accessible to the surgeon [21]. However, the selection of blood vessels as grafts varies according to the circumstances outlined by the American College of Cardiology and the American Heart Association. Table 1 summarizes the mechanical properties reported for the GSV and IMA.

As shown in Table 1 it is possible to identify that the GSV presents better mechanical properties such as burst pressure, suture retention strength, and longitudinal tensile strength compared to IMA. However, other properties such as dynamic compliance, internal diameter, wall thickness, and circumferential tensile strength are lower due to the wall composition in veins presenting some differences compared to the arteries. For example, a lower wall thickness provides a difference in the structure of the veins because of the thinner media layer, lower amount of elastin, and relatively high collagen content [12]. For better comparison, details on the mechanical properties of IMA and GSV are shown in a radar chart in Figure 3. 

## 4. Polytetrafluoroethylene (PTFE) Has the Highest Share on VGs Market

The global vascular graft market is increasing due to the prevalence of cardiovascular diseases related to low levels of physical activity and a sedentary lifestyle. It is expected to reach 3.3 billion by 2026 at a CAGR of 6.4%. Polytetrafluoroethylene (PTFE or Teflon) based vascular grafts are currently the most used. PTFE as a raw material reached 710.3 million USD in 2018, and it is expected to present the fastest CARG over the forecast period due to the graft advantages related to low delamination, minimal blood loss, and excellent mechanical properties [30]. 

PTFE vascular grafts are the nonwoven type. They were developed in 1972, with their first use as a venous prosthesis in a swine model, where they have since been used for more than 30 years. As PTFE-based VG has been the most successful type for small diameter applications; we have considered it a reference for superior mechanical properties.

### 4.1. Micro and Macrostructure of PTFE VGs

PTFE is a thermoplastic and crystalline fluorocarbon polymer like polyethylene; fluoride atoms have substituted hydrogen atoms. These Fluorinated carbons have a high affinity for avoiding reactions with other molecules, for which the PTFE is inert, biocompatible, and avoids thrombogenesis. Implantable PTFE devices were commonly manufactured from a single film of PTFE and subsequently stretched to produce a microporous structure from the non-porous film; the degree of porosity depends on the stretch. PTFE VGs nowadays are produced by extrusion and sintering in a porous tube with fibrils and nodules of controllable sizes. Therefore, the final PTFE VG comprises a fibrous structure defined by interconnected interspaced nodes [31]. Since patency loss results from a lack of tissue regeneration, the internodal distance of PTFE grafts can be modified to enhance tissue ingrowth and endothelization. However, higher porosity reduces the overall circumferential tensile strength and suture retention strength, negatively affecting the handling [32].

### 4.2. Mechanical Properties of PTFE VGs

Despite the wide use of PTFE-based vascular grafts, PTFE is a non-biodegradable biomaterial with low compliance and moderate stiffness. Due to this factor, PTFE VGs have been evolving through different modifications to improve the handling properties and mechanical properties. To improve mechanical properties, especially for lower limb bypass, reconstructions above or below the knee, extra-anatomic procedures, and vascular access grafts, PTFE VGs include internal or external support in the form of rings or spirals offering kink and compression resistance [33]. On the other hand, they have also been coated to improve blood compatibility, avoiding thrombus formation and related patency loss; these coated PTFE VGs are often used for limb reconstruction and vascular access, or for early cannulation vascular access [34,35]. Even multilayered PTFE grafts include an anti-thrombogenic coating and an elastomeric layer intended to mechanically seal the needle entry hole after its removal [32]; PTFE grafts have also been modified to improve longitudinal stretching to adapt to different anatomies. Table 2 summarizes the mean mechanical properties in average commercial PTFE vascular grafts for Arteriovenous Fistula with a 6 mm internal diameter.

As shown in Table 2, PTFE grafts have a high isotropic modulus, whereas the native arteries usually have a lower modulus and are anisotropic. The stiffer feature of the PTFE VGs has been shown to cause a high impedance within the pulsatile flow as the propagation velocity of pressure and flow waves increase. Therefore, in the anastomosis, the reflected waves from pressure and flow transform into different phases that increase the risk of turbulence and unstable flow regions [37]. These forces create a high wall shear stress, ultimately leading to the gene expression related to the development of intimal hyperplasia and atherogenesis. This idea is supported by further compliance reduction of 50% to 86% after 12 weeks of implantation related to foreign body response and fibrous capsule formation [39]. As shown in Figure 4, when comparing the mechanical properties of PTFE vascular grafts with the most successful vascular graft from the great saphenous vein (GSV), it is possible to identify that the compliance is almost two times greater in the GSV. In this sense, Salacinski et al. performed a linear regression analysis comparing the compliance of host arteries, saphenous vein, umbilical veins, bovine xenograft, PET, and PTFE-based grafts with their patency rates at three years of clinical trials. The main results are that the graft patency decreases as the compliance mismatch increases due to the low or absent increase in compliance under the changes in blood pressure due to the lack of viscoelastic properties. Details on the mechanical properties of the PTFE synthetic VG compared with those of the Great Saphenous Vein are shown in Figure 4.

## 5. The Potential of TEVGs on Clinical Success

Even though plenty of development has been reached over the last 50 years in the field of synthetic materials for the design of vascular grafts and favorable results have been obtained for aortic and wide arteries replacement, there have not been any satisfactory results in small caliber grafts due to thrombus formations and poor patency rates [40]. Nevertheless, substantial efforts have been undertaken to overcome these limitations. The objective of developing TEVGs is to create a vascular graft that integrates with the native tissue and behaves like a native vascular vessel, providing the right biochemical and biomechanical stimuli required for the growth and self-regeneration [41].

As stated above, patency loss is a severe complication of small diameter grafts. In this sense, PTFE VGs have been reported to present patency rates of 75% at one year, which will decrease to 44% at five years. This behavior is non-comparable to the saphenous vein with a reported patency rate of 91% at one year, decreasing to 76% at five years, superior to PTFE in all stages. On the other hand, some types of TEVGs show reasonable patency rates. One example is Artegraft^®^, with a mean patency rate of 73% over 18 months. Another example can be the Humacyte graft, with a patency rate of 83% at three months and 60% after six months [42]. Therefore, regenerating the vascular wall through TEVGs represents an opportunity to maintain long-term patency.

### 5.1. Failure Causes of TEVG’s Related to the Mechanical Properties

The leading cause of failure of VG and TEVGs is patency loss where blood flow becomes compromised and can occur through different mechanisms. The most common mechanism is related to thrombogenesis when the surface of the biomaterial lacks the required hemocompatibility [15]. Furthermore, this is the most common cause of failure on VGs and TEVGs used as arteriovenous fistulas, given that clots are formed after repeated puncture, and the thrombus might spread if there is not an anticoagulant surface or treatment. In these cases, secondary patency in a VG is achieved by removing the clot with a catheter, and the lumen of the graft can be restored, allowing blood perfusion. However, this procedure is not always successful if the main thrombogenic conditions are still present, in which case the condition may evolve towards a fibrotic thrombus that will make it impossible to perform a new procedure to recover patency [14].

However, the most common cause of failure related to the mechanical properties of the VG or TEVG includes intimal hyperplasia, in which the overproliferation of smooth muscle cells thickens the tunica intima in the blood vessel, causing the contraction of the construct and the loss of patency [16]. On the other hand, the formation of fibrotic tissue surrounding the graft–in this case, the microstructure of the VG or TEVG–does not allow cell infiltration, and a dense layer of collagen is formed surrounding the graft. In this case, stem cells differentiate toward myofibroblasts, causing the contraction of the VG or TEVG, and stenosis is developed [43]. However, another critical failure cause is the formation of aneurysms in which the chemical or biological degradation of the graft and/or structural defects may lead to the dilatation or rupture of the graft [44].

#### Unstable Flow Conditions and Intimal Hyperplasia Development in TEVGs

Intimal hyperplasia has been reported to present in 10–30% of failure causes of VGs. This high failure rate has been correlated with the compliance mismatch between the VG and the native vessel. In this sense, it has also been reported that the patency loss is directly proportional to the compliance mismatch [45]. The low compliance not only depends on the biomaterial origin but also depends on the microstructure. For instance, it has been reported that there is a strong correlation between low porosity and increasing TEVG wall thickness with a low compliance [46].

The flow stability is key to maintaining the graft patency. The flow regime, velocity profile, and cyclical deformation caused by the pulsatile flow create determinant wall shear stresses (WSS), which are mechanical signals in the cells interacting and repopulating the TEVG [46]. From this perspective, low compliances have also been strongly correlated with low WSS due to the low-compliant grafts presenting a significant difference in diameter under the pulsatile pressure [45]. Furthermore, while the VG maintains a constant diameter, the artery will dilate and contract. When the artery dilates, the VG maintains its low diameter, and the blood flow profile produces a sizeable corresponding effect. This profile is characterized by zones of blood recirculation negatively affecting the velocity profile, inducing turbulent flows, and altering the pressure wave, decreasing the WSS [45]. Figure 5 summarizes the effect of low, medium, and high compliance on the blood flow pattern. 

Regarding the compliance mismatch effect on WSS, normal WSS reported on small diameter vessels—such as the coronary artery—is 0.68 N/m^2^ ranging between 0.3 to 1.24 N/m^2^ [47]. It has been reported that non-compliant VGs present low shear stress near 0.03 N/m^2^, medium compliance VGs display similar shear stresses with an average of 1.04 N/m^2^, and high compliance grafts present higher WSS with an average of 3.6 N/m^2^ [45]. Based on this correlation, it could be expected that low-compliant TEVGs will induce intimal hyperplasia.

Accordingly, TEVGs for different applications must consider the compliance values and the physiological shear stress in which the TEVG will be anastomosed. For instance, GSV has an average compliance value of 4.4%/100 mmHg, and it has been reported that implanted GSV grafts for coronary artery bypass present a range of 1.22 N/m^2^ to 1.73 N/m^2^, whereas GSVs grafts below 0.71 N/m^2^ are predictors for the VG failure [48]. Recent reports have also shown that the left internal mammary artery (IMA), with compliance of 5.22%/100 mmHg, used as a VG for a coronary bypass, maintained its patency for one year and presented high WSS values near 4.43 N/m^2^, contrasted with those occluded with a WSS 2.56 N/m^2^ [49]. Although the differences between the compliance of the GSV and IMA are not significantly different, the WSS produced as VGs for CABG are very different due to differences between the layer’s composition in the media and adventitia layers. 

Although GSV has a different structure from arteries, the vein adapts to the arterial environment due to the different hemodynamic conditions and increased oxygen tension [50]. The most significant change is the increase in the vein diameter between 20% and 85% for arteriovenous fistulas or lower extremity grafts, respectively. The vein wall also augments the wall thickness due to the increased pressure from the proliferation of smooth muscle cells and adventitial fibroblasts derived from bone marrow progenitor cells. These changes have been associated with the normalization of shear stress, occurring during the first months where the initial shear stress will reach up to 9.6 N/m^2^, leading to a patency rate between 75–90% at one year and 50% at 15 years [51].

### 5.2. Biomechanical Stimulation for Physiological Regenerative Responses; Physiological Wall Shear Stresses

When developing TEVGs, it is essential to induce the regeneration of the vascular wall within the anastomosis; not only will the bioactive chemical properties of the graft induce this process, but biomechanical signals are required. Aside from the microstructure allowing the cell infiltration and proliferation, the wall shear stress will generate different responses in the cells interacting with the biomaterial in the hemodynamic context. 

Therefore, physiological WSS will lead to the regeneration of the vascular wall, whereas the low WSS induced by the lack of compliance will have a detrimental effect on the gene expression of immune cells, smooth muscle cells, and endothelial cells. For instance, for regeneration to occur, the inflammatory response must be modulated towards the differentiation of Macrophages to the M2 type, recognized by the cytokine secretion to induce extracellular matrix deposition. Low WSS has been correlated to the maintenance of the M1 phenotype leading to chronic inflammatory responses.

On the other hand, smooth muscle cells must begin their differentiation towards a contractile phenotype rather than a synthetic phenotype. The synthetic phenotype related to low WSS has been associated with lower blood pressure responses affecting the construct’s overall compliance. Finally, endothelial cells should differentiate towards a functional phenotype able to release nitric oxide (NO) as a regulator of the vascular wall tone. Low WSS has been shown to induce senescence on endothelial cells without proliferation capacity and limited release of NO. Table 3 summarizes the effect of baseline and lows WSS on the endothelial cells, smooth muscle cells, and macrophage gene expression. 

According to the data registered in Table 3, it can be observed that the low WSS generates the down-regulation in the expression of eNOS (endothelial nitric oxide synthase), the enzyme responsible for the NO release and the maintenance of the overall homeostasis in the vascular wall [52]. In addition, it generates the down-regulation of NOTCH1 (Neurogenic locus notch homolog protein 1), which is a mechanical sensor that maintains the junctional integrity of endothelium [53], and NOX 4 (NADPH oxidase 4), which produces H_2_O_2_ as a signaling molecule for endothelial cell proliferation [54]. Without the expression of these genes, it is likely improbable that the endothelial lining can be regenerated over the TEVG surface. 

On the other hand, low WSS on endothelial cells also increases the expression of MCP-1 (Monocyte Chemoattractant Protein-1), which is a chemoattractant for proinflammatory monocytes. At the same time, the expression of VCAM-1 (Vascular cell adhesion protein-1), ICAM-1 (Intercellular Adhesion Molecule-1), and EDN-1 (Endothelin-1) increases, which are adhesive molecules for monocytes, and PDGF (platelet-derived growth factor) also increases, promoting thrombus formation. 

Regarding smooth muscle cells, a low WSS decreases the expression of a-SMA (smooth muscle actin), SM22 (Transgelin), SMTN (Smothelin), and CNN (calponin); all are genes related to the contractile function of smooth muscle cells required to respond to contractile and dilating signals from endothelial cells [16]. Furthermore, low WSS induces the up-regulation of proliferative genes, responsible for intimal hyperplasia, TGF-β1 (transforming growth factor-beta) inducing inflammation, and the MMP2 (matrix metalloproteinase-2) that not only degrades the vascular wall but also increase the migration and proliferation of the smooth muscle cells with the synthetic phenotype [51,55]. 

Finally, macrophage modulation due to low WSS has been shown to reduce the expression of M2 phenotype-related genes such as CD206 (macrophage mannose receptor 1), IL-10 (Interleukin 10), which blocks the NF-κB (nuclear factor kappa-light-chain-enhancer of activated B cells) proinflammatory pathway, and TGF-B1. Furthermore, genes related to increased inflammation are reported to be activated, such as the signaling pathway related to NF-κB activation, MCP-1, and Selectin as chemotactic for new inflammatory cells, as well as present an increase in MMP9, related to the degradation of the scaffold [56,57]. 

## 6. Biomechanical Design Requirement for Vascular Grafts

Due to the significant differences between vascular grafts and native arteries in their mechanical properties and behavior under implantation conditions, several requirements have been studied to develop mechanical characterizations of these new grafts. The requirements to evaluate vascular grafts are outlined in International Standards ANSI/AAMI VP20: 1994 (American National Standard for cardiovascular implants and vascular prostheses) [58], ISO 7198:2016 (Cardiovascular implants and extracorporeal systems—Vascular prostheses—Tubular vascular grafts and vascular patches 94, and ASTM F3225-17 (Standard Guide for Characterization and Assessment of Vascular Graft Tissue Engineered Medical Products (TEMPs) [59].

### ANSI/AAMI VP20: 1994, ISO 7198:2016 and ASTM F3225-17

Test guidelines for international standards are defined for all vascular grafts: synthetic, biological, and coated [59]. According to ASTM F3225-17, mechanical testing must be performed in an environment that emulates the vascular grafting conditions of use, most commonly in buffered saline solutions at 37 °C. Otherwise, non-physiological test conditions must be justified. In addition, parameters such as the time between tissue collection and testing and sample storage may modify the mechanical properties [59]. A minimum of three samples from at least three manufactured lots is necessary to have an appropriate variability of the characteristics in the samples studied.

Due to the anisotropic properties associated with vascular grafts, strength tests should be performed, including more than one axis. Therefore, longitudinal, and circumferential tensile strength tests are performed to determine whether the axial and radial yield and/or breakpoint are reached, respectively. Stress vs. strain plots must be made to include and analyze the data. In addition, suture retention strength tests should be developed to determine the force required to pull a suture from the vascular graft to simulate clinical techniques (straight-across, oblique, and longitudinal procedures). The suture is usually placed 2 mm from the edge and tested in various directions to determine the strength that the prosthesis withstands before mechanical failure.

Moreover, a burst strength test is performed to determine the pressure rate change until sample bursting occurs. This method allows to report the diameter of the sample when it is pressurized directly with fluid or gas. A repeated puncture test with a dialysis needle (16G) should be performed to measure the strength that supports the prosthesis through a force test such as pressurized burst strength or circumferential tensile strength [59].

It is necessary to perform tests to determine the thickness of the wall and the relaxed and pressurized internal diameter to observe changes in diameter under different hemodynamic conditions. Furthermore, the vascular graft discontinuity should be reported when the lumen diameter decreases during kinking and the radius of curvature that impedes normal flow through the graft. Another way to evaluate conditions that approach the preclinical environment is by measuring diameter change simulated under cardiac cycle conditions to determine the radial dynamic compliance. In addition, the rate of water leakage through the prosthesis wall should be characterized to avoid leakage at the time of implantation [59].

## 7. Trends on TEVGs Design, Biomaterials, and Manufacture Techniques in the Context of Desired Mechanical Properties

The ideal graft must have comparable mechanical properties to those of the native vessels that will be anastomosed. For instance, it has been shown that the gold standard for the repair of peripheral vessels is the GSV, whereas in other applications such as in Coronary Artery Bypass Graft Surgery (CABG), the gold standard corresponds to the IMA. This is implemented to reduce the level of mechanical decoupling and levels of failure that can lead to stenosis or aneurysms and, therefore, long-term permeability loss. The most important factors involve mechanical strength and compliance concerning the viscoelasticity of the scaffold. Likewise, the grafts must have sufficient mechanical strength and compliance to withstand the changes in blood pressure. At the same time, they should adjust correctly to the adjacent vessels when completing the suture procedure, enabling the correct velocity profiles and continuity in the pressure waves. In the following sections, the current trends in TEVGs design, biomaterials, and manufacturing techniques are reviewed according to the TEVGs that have been reached, including clinical trials, pre-clinical studies on animal models, and techniques that have been tested on in vitro conditions during the last five years. 

### 7.1. TEVGs That Have Reached Clinical Trials

After analyzing the current trends in TEVGs, five clinical trials were identified regarding TEGVs for human use in different contexts. The main mechanical properties of each vascular graft compared to GSV and IMA grafts are summarized in Appendix A. The mean values of the mechanical properties of the TEVGs on clinical trial are summarized and compared in Table 4 and Figure 6. 

The most recent TEVG that has entered to clinical trial phase is a Polyhedral oligomeric silsesquioxane poly (carbonate-urea) urethane (POSS-PCU) small-diameter vascular graft, fabricated through extrusion and phase inversion method using sodium bicarbonate as a porogen. This TEVG is currently being tested as arteriovenous fistulas for hemodialysis; the study began in 2021 and is expected to reach completion in April of 2025. This TEVG has a small diameter (<5 mm) with a mean wall thickness of 0.94 mm that is placed in adult female patients undergoing hemodialysis and in need of new arteriovenous access but without any viable access for dialysis. This study has an inclusion criteria of patients on oral contraceptives or with an intrauterine device, aiming to demonstrate patency in patients with increased risk of thrombosis. It is expected to measure patency rates at 18 months by Doppler ultrasonography (d-US) and compare it to the patency of the PTFE grafts. They also will measure the occurrence of any Serious Adverse Event (SAE) related to the implantation of the POSS-PCU TEVG. For the initial characterization of the POSS-PCU vascular graft, they used Delfino’s strain energy potential to capture the viscoelastic properties of the materials. Later, they conducted different experimental and computational experiments to show shear stress within the graft wall in the context of increased uniaxial strength. They concluded that besides the overall resistance of the graft, the stiffness and long-term viscoelastic properties could be improved with better manufacturing techniques; detailed mechanical properties of this TEVG can be found in the Appendix A [60]. The POSS-PCU TEVG offers similar longitudinal tensile strength compared to GSV and IMA grafts (3210 vs. 2405 and 4300 KPa), however, its suture resistance strength is markedly higher (4460 vs. 3200 and 1350 KPa, respectively). Furthermore, it presents reduced dynamic compliance compared to the allografts (1.59 vs. 4.4 and 5.22%/100 mmHg respectively) may explain its increased stiffness and poor vascular physiology resemblance (Figure 6).

A Human Acellular Vessel (HAV) from Humacyte is a graft also developed for patients with need of hemodialysis access but who are not candidates for a fistula. This TEVG is created in vitro through the culture of human smooth muscle cells and fibroblast cultured over a biodegradable polymer. Further decellularization process creates a structure that retains the extracellular matrix protein and provides corresponding mechanical properties. Animal models were performed in baboons as arteriovenous grafts with 80% patency at six months. This vascular graft has been included in different clinical trials. One of the most relevant, made in 2016 as a phase 2 trial in 40 elected patients, demonstrated primary patency rates of 63% (95% CI 47–72) at six months and 28% (95% CI 17–40) at 12 months. Conversely, secondary assisted patency rates at six and 12 months were 97% (95% CI 85–98) and 89% (95% CI 74–93), respectively. Secondary patency rates in extensive multicenter cohort studies with PTFE were 55–66% at one year, making HAV a suitable option. In the group of patients, they also measured serious adverse events (SAEs) that occurred 155 times in 33 patients, leading to increased reintervention rates. SAEs mainly included patency loss, infection, and one case of steal syndrome. HAV provides an exciting conclusion, as they have better secondary patency rates than PTFE-based VGs, leading to a well-tolerated TEVG with no signs of aneurysm formation or degradation with low immunogenicity. For which it has been included in different clinical trials from 2015 to 2020. Research-ers found that Artegraft® has primary, primary-assisted, and secondary patency rates of immunogenicity. However, HAV has the lower reported tensile strengths (circumferential and longitudinal) along all TEVGs in clinical trial phases (1400 and 1200 KPa) and low dynamic compliance compared to human allografts (1.5 vs. 4.4 and 5.22%/100 mmHg), leading to increased stiffness and a poor vascular physiology resemblance; complete data can be found in the Appendix A.

Likewise, the TRUE vascular graft is also a decellularized arteriovenous TEVG designed by neonatal human dermal fibroblasts seeded in a bovine fibrin gel. A baboon pre-clinical model was performed on 10 subjects to determine its safety in terms of occurrence of adverse events and to find patency rates. They found patency rates at 3 and 6 months of 83% (5 of 6) and 60% (3 of 5), respectively, with evidence of recellularization of smooth muscle cells and endothelium formation. This TEVG is currently being tested as an arteriovenous graft for hemodialysis access on 10 participants beginning in May of 2021 and it is expected to finish in June of 2022. The mechanical properties of this kind of graft can be seen in the Appendix A. However, circumferential tensile strength and suture resistance strength were like the IMA (3800 and 1950 vs. 4100 and 1350 KPa, respectively), making it a newly suitable option for further study.

Another kind of TEVG for hemodialysis accesses is Artegraft^®^, a decellularized bovine carotid artery used for arteriovenous fistula generation. These TEVG have been included in different clinical trials from 2015 to 2020. Researchers found that Artegraft^®^ has primary, primary-assisted, and secondary patency rates of 73.3%, 67%, and 89%, respectively. SAE was present in one immunocompromised patient that presented a resistant infection, leading to the early removal of Artegaft^®^ at two months. A relevant finding is that anastomotic venous stenosis occurred in some grafts and was the most common indication of graft removal. Artegraft^®^ demonstrated similar tensile strength properties compared to PTFE VGs. However, its low dynamic compliance [1,5] can explain the occurrence of stenotic venous anastomosis in most grafts (Figure 6).

BioIntegral Surgical No-React ^®^ are bovine pericardial xenografts used as a strategy for vascular graft infections due to their regenerative properties; these series of clinical trials began in 2019. A prospective study of six patients with infected aortoiliac segments treated with BioIntegral Surgical No-React ^®^ demonstrated that four out of the six patients are still alive with complete patency demonstrated by d-US. Unfortunately, two died from acute myocardial infarction, and the other due to sepsis secondary to the vascular infection. Although this graft has a remarkably increased longitudinal tensile strength compared to GSVs and IMA (10,000 vs. 2405 and 4300 KPa), it conferred to most patients (4 out of 6) a proper control of the infection with reasonable long-term patency rates.

According to the reported data, a mean value was extracted from the mechanical properties of the current vascular grafts undergoing on clinical trials and were compared to the mechanical properties of the GSV, as shown in Table 4 and Figure 6; individual data are shown on a bar graph in Figure 6. 

**Table 4 polymers-14-03440-t004:** Mechanical properties in blood vessels: grafts in clinical trials (CT grafts) and great saphenous vein (GSV).

Test Performed	CT Grafts	GSV	Reference
Internal Diameter (mm)	5	3	[22,23,61,62,63,64]
Wall Thickness (µM)	690	518	[24,25,61,62,63,64,65]
Circumferential Tensile Strength (KPa)	2380	2405	[12,19,26,62,63,64,65]
Longitudinal Tensile Strength (KPa)	4230	9760	[61,62,63,65]
Burst Pressure (KPa)	405.17	371.96	[12,26,62,64]
Suture Retention Strength (g)	306	327	[22,27,61,62,64]
Dynamic Compliance (%/100 mmHg)	1.53	4.40	[12,28,61,62,63]

### 7.2. TEVGs That Have Reached Pre-Clinical Animal Models

After the analysis of the recruited data, a total of 43 studies fulfilled inclusion criteria for TEVGs that have reached pre-clinical animal models [66,67,68,69,70,71,72,73,74,75,76,77,78,79,80,81,82,83,84,85,86,87,88,89,90,91,92,93,94,95,96,97,98,99,100,101,102,103,104,105,106,107]. It was found that 46.43% (*n* = 26; 46.46%) of the studies reported the use of Poly ε-caprolactone (PCL), mostly manufactured through electrospinning (*n* = 23, 88.46%) and more frequently tested on murine animal models (rats). PCL has been shown to be biocompatible, and the constructs exhibit slow degradation rates of about 1–2 years [67,68,70,72,74,76,78,79,80,81,82,89,90,92,95,96,97,99,102,103,104,105,107]. This has been correlated with the ability to maintain the TEVGs stability under hemodynamic operative conditions and less acidic breakdown products compared to other polyesters because it can be easily degraded by lipases and macrophages, presenting a low inflammatory profile and the potential for loadbearing [29]. However, although it has been proven that PCL is a relatively easy material to work with, PCL-Based TEVGs usually present low compliance compared to native vessels (GSV). From this perspective, the raw PCL has a compliance value close to 2%/mmHg, and PCL-Based TEVGs present a value close to 3%/mmHg; both values are lower than those of the GSV or IMA, closer to 5%/mmHg.

To overcome these limitations, various authors have chosen to use a variation of PCL such as poly (L-lactide-co-ε-caprolactone) or PLCL with a softer and more elastic nature. Compared to PCL, PLCL-based TEVGs present higher compliance values close to 8%/mmHg [108]. It has also been reported that PLCL has excellent mechanical properties such as high plasticity and higher degradation rates than PCL. Nevertheless, the lactide group inclusion generates low biocompatibility, poor hydrophilicity, and presents acidic degradation.

Polyurethane-Based TEVGs (PUs) are the second most reported TEVGs (*n* = 6; 10.71%). PUs has been considered as a good base biomaterial due to its relatively high tensile and flexural strength [37,77,81,91,93,100,104]. Furthermore, it has been shown that PU has relatively high compliance compared to other materials. PU-Based TEVGs present an average compliance value of 6.5%/mmHg [37], being closer to the value found in native arteries. However, although none of the consulted manuscripts had a specific value for the circumferential tensile strength of the material, it has been reported that repeated puncture of PU-Based TEVGs might lead to aneurism generation, meaning that the material’s circumferential strength might be compromised, and it could thus not be comparable to the native arteries. Therefore, for its use in TEVGs, different alternatives should be included to improve the circumferential strength of the material while maintaining the required elasticity for compliance.

The third most common biomaterial for TEVGs fabrication corresponds to decellularized arteries (*n* = 4; 7.14%), which can be correlated to the advantages of maintaining the structure and mechanical properties of native tissues. Moreover, these biomaterials have a relatively low immune response from the patient, cause minor damage to other bioactive components, and usually do not use chemical reagents to reduce adverse reactions in the body [109]. Regarding the compliance of the material, it has been reported that the compliance of this type of graft is close to 9.7%/mmHg, which is a high value compared to previously described materials. However, it has also been reported that most decellularized grafts involve high costs, and they can also require two or more surgeries.

All these data are summarized in Table 5 and Figure 7, comparing the mechanical properties the average TEVG entered to pre-clinical models with GSV, this is a PCL-Based TEVG and fabricated with Electrospinning. Appendix A are summarized in Appendix A. 

### 7.3. Current Strategies of TEVGs on In Vitro Testing 

To succeed with international standards, various authors have overseen the innovation of new techniques, materials, and practices for developing vascular grafts. A total of 46 articles fulfilled the inclusion criteria regarding TEVGs undergoing in vitro testing. Among the most used materials in in vitro processes, the PCL, TPU (Thermoplastic Polyurethane), and PLGA (poly(lactic-co-glycolic acid)) are the most used, together with natural biomaterials such as collagen and gelatin [109,110,111,112,113,114,115,116,117,118,119,120,121,122,123,124,125,126,127,128,129,130,131,132,133,134,135,136,137,138,139,140,141,142,143,144,145,146,147,148,149,150,151,152,153,154,155,156]. 

Based on these biomaterials, there has been a significant trend of various manufacturing methods, in which the electrospinning method is one of the most used. This methodology is based on the formation of nanofibers through the application of electricity to the material. This process is usually used due to its versatility in adjusting different parameters such as porosity, size, thickness, and density of the graft. About 37% of the articles mentioned used the electrospinning method in manufacturing [114,116,118,120,121,122,123,125,127,128,132,133,134,135,137,139,141,142,143,144,146,147,150,151,154,155]. On the other hand, the second most used process is the Solvent Casting method, with 13.04% of articles using this process [117,136,148,149,156]. 

However, different authors carry out mechanical and cell viability tests to prove the possible behavior of the grafts in a patient. Due to the mechanical tests, a trend has been found regarding the tests carried out, of which the longitudinal tensile strength tests are most often reported. This measurement helps to determine the vascular graft resistance against longitudinal tensile forces. Along with the revised manuscripts, an average of 94,300 KPa (*n* = 37) was evidenced, thus indicating that most of the grafts resist high longitudinal forces [113,114,115,116,117,118,119,120,121,122,123,124,125,126,127,128,129,130,131,132,133,134,135,136,137,138,139,140,141,142,143,144,145,146,147,148,149,150,151,152,153,154,155,156]. Nonetheless, circumferential tensile strength is not often reported, same as compliance, both being highly important data required to establish suitable mechanical properties for TEVGs applications. On the other hand, the second most reported measurement is the Burst Strength since it is essential to determine the pressure that can resist substantial flows. The authors recorded a mean value of 2161.14 mmHg (*n* = 17) [118,120,122,123,126,128,142,143,146,147,151,152,154,155,156,157].

Likewise, some articles have also included other relevant tests to comply with international standards. Another of the most important tests is circumferential traction resistance, with an average result of 9210 KPa (*n* = 13), suture retention resistance with an average result of 7.34 N (*n* = 11), and finally dynamic compliance with a mean result of 327.28 mmHg (*n* = 6). This can be evidenced in Figure 7 and Figure 8, where it is also possible to observe the morphology of the synthetic vascular grafts compared to the great saphenous vein. Regarding the radar chart, it is possible to analyze the role of PCL in the mechanical properties provided to the grafts, where it stands out over TPU and PLGA. These materials show different factors such as longitudinal tensile strength, circumferential tensile strength, suture retention strength, and dynamic compliance, thus making it a material with the necessary characteristics of durability and hardness for a device that is going to be subjected to such a changing environment. It was previously mentioned that the ideal graft should have mechanical characteristics similar to a vein. In this way, it can be argued that the PCL meets these characteristics, even surpassing the behavior of the superior saphenous vein in said properties. For these reasons, there has been a trend among authors to use this material by combining it with one that can help improve properties such as burst pressure, and thus generate an ideal replacement material [120,123,124,125,127,132,133,135,139,140,142,145,147,149,154,155,156,157].

Another critical aspect of in vitro tests is cell analysis. Cell viability, cytotoxicity, and cell adhesion, among others, are necessary tests to comply with standards. Thus, the researchers have developed several tests to reach a physiological environment and understand what the possible behavior of the graft will be in an authentic context.

This review has found interesting data regarding the biomaterials used for TEVGs that have been reported in in vitro testing (Figure 8). The three most common materials in manufacturing TEGVs (Appendix A) and tested in vitro were PCL (37.5%), TPU (8.93%), and PLGA (7.14%). PCL-based TEGVs tend to be more like the internal mammary artery in circumferential (3594 ± 2486 vs. 4100 KPa), longitudinal (4107 ± 2376 vs. 4300 KPa) tensile strengths, and dynamic compliance (5.42 ± 2.65 vs. 5.22%/100 mmHg), respectively. However, its suture resistance strength (4520 ± 4130 KPa) is much higher than that of saphenous vein grafts and the internal mammary artery (3200 and 1350 KPa), respectively, leading to decreased adaptability of the TEVG to suture tension. In addition, 12 out of 22 TEGVs in vitro studies reported the presence of endothelization of the vascular graft lumen. The principal manufacturing technique was electrospinning (16 out of 22) and most products were monolayered (11 out of 22) [118,127,132,137,139,140,141,145,147,154,155,156,157].

On the other hand, TPU-based TEGVs showed the highest mechanical properties regarding longitudinal tensile strength (15,750 ± 4546 KPa), suture resistance strength (7860 ± 1600 KPa) compared to saphenous vein grafts (2405 and 3200 KPa), and internal mammary artery (4300 and 1300 KPa), respectively. These results show that TPU is a highly rigid material that does not resemble vascular physiology. However, when used as part of a mixture of other materials, TPU can provide the rigidity necessary for more flexible materials rather than being used alone. Regardless of there being no information regarding dynamic compliance, three out of five TPU-based TEGVs had endothelization. All TEGVs were manufactured by electrospinning [118,137,141,154].

Finally, PLGA-based TEGVs had intermediate mechanical properties. Its circumferential tensile strength (8350 KPa) and suture resistance strength (1950 ± 1670 KPa) was higher than the internal mammary artery properties (4100 and 1350 KPa), but lower than the saphenous vein graft (9760 and 3200 KPa), respectively. Its longitudinal tensile strength (5008 ± 3468 KPa) was higher than the saphenous vein graft (2450 KPa) and the internal mammary artery (4300 KPa), but its dynamic compliance (3.41%/100 mmHg) was lower than both vessels (4.4 and 5.22%/100 mmHg). In addition, endothelization was found in two out of four PLGA-based TEGVs and the principal manufacturing technique was electrospinning. These results conclude that PLGA is a rigid material that, as with TPU, can give rigidity to a TEVG, but more flexible materials need to be used to offer compliance to the vascular graft [118,119,137,141,154]. 

All these data are summarized in Table 6 and Figure 8, comparing the mechanical properties average most common TEVGs from in vitro studies (PCL, TPU, and PLGA) with the GSV. Complete data is summarized in Appendix A.

### 7.4. Promising Biomaterials and Clinical Practice Correlation on TEVGs Applications

Despite the advances made in TEVGs development, the pursuit of the optimal mechanical properties that approximate those of native arteries is still ongoing. Multiple factors are essential to achieve suitable surgical and mechanical properties, including the proper selection of biomaterials, manufacturing techniques, and surface functionalizations. For instance, new biomaterials have been developed to increase patency rates and avoid complications such as pulmonary thromboembolism [158]. In these sections, we are only talking about newer approaches that have been proposed for use on TEVGs development. 

The article published by Bai et al. showed structured scaffolds with biodegradable polyester-polydepsipeptid and silk fibroin (PCL-PIBMD/SF). The manufacturing process was on a sandwich-like composite delivering plasmid complexes aiming to induce endothelialization and was manufactured through layer-by-layer electrospinning and electrospraying techniques. The scaffold was tested with human umbilical vein endothelial cell cultures (HUVECs). The average diameter of nanofibers decreased from 573.8 nm to 285.1 nm, with SF content increasing from 0 to 90%. The mechanical properties found were suitable for vascular scaffolds when the weight ratio of SF was 10%, as the found tensile strength and elongation were 7050.34 kPa and 210 ± 21%. A porosity of 27.5 ± 7.4% with a weight ratio of 90/10 was reported. The HUVECs coverage ratio on day three was 66 ± 3%. This scaffold showed mechanical properties that suggest promising application for TEVGs and also demonstrated the promotion of cell proliferation, adhesion, spreading, and migration [159].

Adding to the research, the study published by Xie et al. designed a tissue-engineered vascular scaffold with portulaca flavonoid (PTF) via electrospinning and was integrated with PCL. It was tested on a culture of human vascular smooth muscle cells (HVSMCs). The scaffold exhibited biomimetic net-like fiber structures, an elastic modulus of 2–20 MPa, the ultimate tensile stress of 2000 kPa, and a fracture strain of 60% in the transverse direction. In addition, it showed that compared to the PCL scaffold, the integration of bioactive PTF had better hydrophilicity and degradability. In addition, inhibition of abnormal intimal hyperplasia was observed [160].

Moreover, one of the most critical factors in understanding the clinical use of each native artery/vein is the clinical practice guidelines that continue to support using native arteries or veins. However, we have observed that native arteries/veins are used for the clinical condition and the patient’s requirements, whereas even for the same pathology, different vessels with very different mechanical properties are used. In pathologies involving small vessels, the evidence supports the individualized and specific use of different types of native vessels. The clearest example is the variability between myocardial revascularization per se and infrainguinal revascularization.

The ACC/AHA/SCAI Coronary Revascularization Guidelines published in 2021 recommend using the radial artery in isolated Coronary Artery Bypass Grafting (CABG) and the left mammary artery in multiple bypasses over the saphenous vein. The evidence shows that using the left mammary artery prolongs the survival of the patient who only requires isolated CABG in a significantly stenosed non–Left Anterior Descending (LAD) vessel. In addition, the case of the radial artery has shown that in the medium and long term, it has higher patency rates and better clinical results at ten years of undergoing CABG to bypass the LAD [161].

On the other hand, the Society for Vascular Surgery practice guidelines for the atherosclerotic occlusive disease of the lower extremities recommends infrainguinal bypass using the saphenous vein [162].

Depending on the patient’s need and underlying pathology, this variability in mechanical property requirements also applies to vascular grafts. Thus, evaluating the different mechanical properties in each clinical indication is necessary since it is possible to hypothesize that not only a single small vascular graft with specific mechanical properties will be required.

### 7.5. Approaches/Techniques for Fabrication of Small Diameter TEVGs

Currently, there is a great diversity of manufacturing techniques for small diameter TEVGs. Several biomaterials have the plasticity required for application on different manufacturing techniques. Most studies reported the use of electrospinning as the primary method of fabrication (*n* = 25; 58.14%); this is because it offers the ability to fine-tune mechanical properties during the fabrication processes while also offering precise control over the composition, dimension, and alignment of the fiber of the material [66,69,70,72,78,80,81,82,83,84,88,93,95,96,97,99,100,101,103,104,107,110]. Furthermore, it can combine synthetic and natural materials, meeting specific needs like high mechanical durability in terms of high burst strength and compliance. Finally, this method allows the incorporation of natural polymers that promote the proliferation of different cell types in the matrix of the graft’s wall.

Although electrospinning as the primary manufacturing technique is one of the most common for the development of grafts, the fiber diameters should be considered. The reasoning is that if the diameter is too small, the TEVG will have low porosity, meaning that cell infiltration will be limited. On the other hand, if the diameter of the fiber is too big, the biomechanical graft properties could be compromised due to an increase in the porosity, leading to blood leakages. It has been reported that thicker-fiber grafts (5–6 μm) tend to polarize into the immunomodulatory and tissue remodeling (M2) phenotype, while thinner-fiber grafts (2–3 μm) express a pro-inflammatory (M1) phenotype [111]. In this case, the average fiber diameter reported for the PCL TEVGs data was 2.4, indicating that this diameter should be improved to enhance regenerative outcomes.

Another relatively common technique typically used for the manufacture of TEVGs is Freeze tawing (*n* = 3; 6.98%) [67,76,86,163,164,165]; some of the advantages of this method are that the structure of the material is maintained, moisture is removed at low temperatures, the stability of the material is increased during storage, and the fast transition of the moisturized material to be dehydrated minimized several degradation reactions [112]. Regarding the use of this method in the manufacture of TEVGs, results have reported that the porosity of the graft is high enough to polarize into the immunomodulatory and tissue remodeling (M2) phenotype, meaning that there should be an improvement in regenerative outcome. Nevertheless, this technique has not been widely used now that some mechanical properties seem to be affected. For example, the circumferential tensile strength seems to be lowered compared to other methods such as electrospinning.

## 8. Biomechanical Properties of Surfaces and Vascular Infections on TEVGs

Another topic related to the physicochemical properties of TEVGs and its mechanical properties are the surfaces of the designed scaffolds. Depending on these properties, biomechanical stimuli–the same as the effect of WSS–might have an impact on cell adhesion and regeneration, or even might promote bacterial colonization. For instance, some micropatterns as well as stiffer surfaces promote cell adhesion and migration, and these data have been widely reported. 

Nevertheless, one of the least studied topics on TEVGs is how to prevent vascular infections, which is one of the most detrimental complications. Due to the increase in antimicrobial resistance and infection of different types of implants, multiple strategies have been studied to prevent bacterial colonization and biofilm formation. One of the most efficient strategies is a colonization-resistant surface by coating the surface with the inclusion of superhydrophobic, uncharged, or highly hydrophilic molecules that do not allow bacterial adhesion. For example, polyethylene glycol and zwitterionic polymers have been shown to inhibit bacterial adhesion because they generate a stearic repulsion and movement phenomenon. Hydrophilic molecules such as heparin have also been used to inhibit bacterial adhesion. Superhydrophobic structures with specific topographies have shown efficiency in decreasing adhesion [166,167,168].

Although less studied, microparticles and nanoparticles with antimicrobial peptides are also used for functionalization, with a wide range of inhibitory effects against bacteria, fungi, parasites, and viruses. Other alternatives include coatings with silver, aluminum, cobalt, zinc, and copper, given that the ions destabilize the bacterial membrane. The same principle applies to Cationic polymers such as chitosan and polyethyleneimine (PEI). Other strategies include local mechanisms of controlled release of antibiotics such as gentamicin, amoxicillin, and vancomycin, among others. For this purpose, multiple transport media such as alginate nanoparticles, hydrogels loaded with these molecules, and even mixing biomaterials with these antibiotics have been employed [169,170,171,172,173].

A 2003 meta-analysis found that autologous vein repairs at any site have a better prognosis and better long-term outcomes regarding graft patency than PTFE grafts in peripheral vessels. In addition, evidence regarding the use of other materials such as bovine pericardium in abdominal aortic repairs, and rifampicin-embedded or silver-embedded prostheses has come to be considered. However, reinfection rates are often higher than 10%. So far, the only regenerative vascular graft for application in aortic aneurysm repair and large-diameter peripheral vessel repair is Omniflow II from the commercial house LeMaitré. Omniflow II is a biosynthetic vascular graft composed of a polyester mesh as an endoskeleton covered by cross-linked ovine collagen. Its simple fabrication is based on the immune response to foreign body and fibrous tissue formation on the polyester mesh implanted in the animal after a few months. 

Collagen-based biosynthetic vascular grafts have demonstrated graft patency rates approaching 70–80% at one year, with favorable rates (57–64%) in below-knee reconstructions. Infection rates are meager, being lower than prosthetic grafts due to better healing patterns linked to their collagen structure, better integration with host tissue, and evidence of capillary growth or micro vascularization of the graft. In the diabetic population–who are at higher risk of infection–low incidences of reinfection have been found, ranging from 2.8 to 4.8%–much lower than in other types of materials. In this regard, the evidence associated with collagen biosynthetic grafts tends to be very promising. However, their use has only been described in case reports and very small cohorts, especially in repairs associated with the abdominal aorta, which is why more evidence is needed. For this reason, a regenerative vascular graft that reduces reoperation rates and ensures laminar flow through the anastomoses will guarantee long-term graft patency and limb salvage [173,174,175]. 

Various biodegradable biomaterials with bacteriostatic/bactericidal activity have been developed. For this purpose, electrospinning–an additive manufacturing technique for the controlled deposition of nanofibers–has been one of the most widely used techniques. Using antibacterial agents such as tetracycline, chlorhexidine, triclosan, and even eugenol loaded on fibers obtained by electrospinning biodegradable polymers as a wound dressing in tissue engineering has been reported. In 2018, Zhenguang, L et al. designed membranes by electrospinning polycaprolactone (PCL) and gelatin incorporating eugenol and adhesive peptides for endothelial cells as a method for fabricating a regenerative vascular graft with bactericidal properties. Its inhibitory antimicrobial activity against S.aureus was 78% and against E.coli was 74%, which was attributed to the ability to destabilize the cell membrane [135,176,177,178]. 

As for bacteriostatic/bactericidal vascular grafts, none are currently on the market with this indication, and the literature reports few approaches. However, one of the most common is the fabrication of vascular grafts with hydrogels or polymers, including chitosan. Due to their bacteriostatic properties, they have helped stop the microbial activity. In addition, some of these TEVGs include other molecules such as silver and even heparin, which, although their general purpose is not to contribute to antimicrobial activity, do have this function.

## 9. Conclusions

This literature review aimed to identify the different trends in the use of materials to develop readily available TEVGs without pre-seeded cells. We show that PCL is currently one of the most used polymers in both in vivo and in vitro studies. However, lack of compliance with PCL-based TEVG has been identified as one of the main limitations of its long-term implementation in clinical trials. Other polymers such as TPU, PU, and PLGA show relevant properties to offer the required circumferential and longitudinal tensile strength for TEVGs applications. However, their use should be carefully implemented, including other materials to offer the required flexibility required for improved compliance and to provide the biomechanical properties suitable for TEVGS applications.

On the other hand, electrospinning is currently the most used manufacturing method, offering standardization and industrialization of the TEVGs. Furthermore, it is easy to use for various polymers while providing proper microstructure and macrostructure. Nonetheless, more studies are still required in manufacturing to reach an ideal fiber thickness and porosity for the grafts since cell migration can be affected while maintaining the long-term graft mechanical properties. 

Strategies such as decellularized blood vessels offer a suitable alternative for TEVGs since their mechanical properties are very similar to the native tissues and are successful in the regeneration process. However, standardization and industrialization are more complex and will require multiple steps to be commercialized. 

Personalized medicine marks the future of health sciences, and the use of vascular grafts from tissue engineering allows us not only to create alternatives for each of the vascular pathologies but also to offer quality and durable elements that meet each patient’s requirements concerning their vascular physiology and rheology. 

Finally, the biomaterials and manufacture methods should ensure the compliance of the anastomosed artery or blood vessel, considering that the increase in the compliance mismatch is directly correlated to the long-term patency loss due to the low wall shear stresses generated by the flow pattern disturbances along the blood vessel and the vascular graft. Therefore, the main objective of a TEVG to provide the required biomechanical signals for regeneration is to reach proper compliance. 

## Figures and Tables

**Figure 1 polymers-14-03440-f001:**
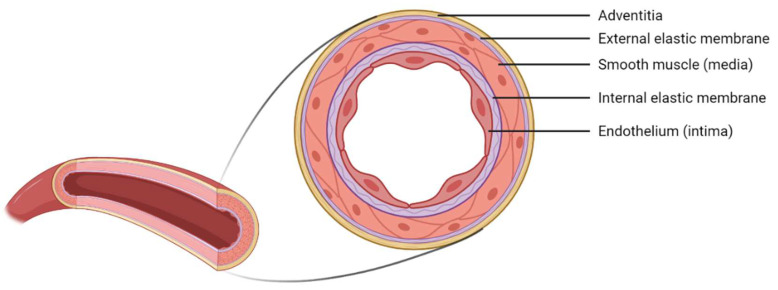
Structure and layers of blood vessels. Graph created with BioRender.com (accessed on 29 May 2022).

**Figure 2 polymers-14-03440-f002:**
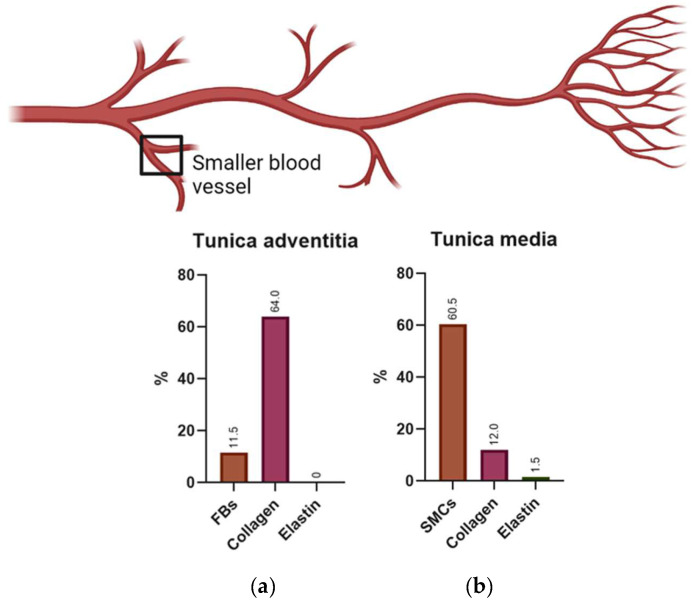
Component percentages on small-sized blood vessels for (**a**) tunica adventitia and (**b**) tunica media. Data from small-sized vessels retrieved from D.B. Camasão et al. [12]. Fibroblasts (FBs) and smooth muscle cells (SMCs). Image created with BioRender.com and graph constructed with GraphPad Prism (accessed on 29 May 2022).

**Figure 3 polymers-14-03440-f003:**
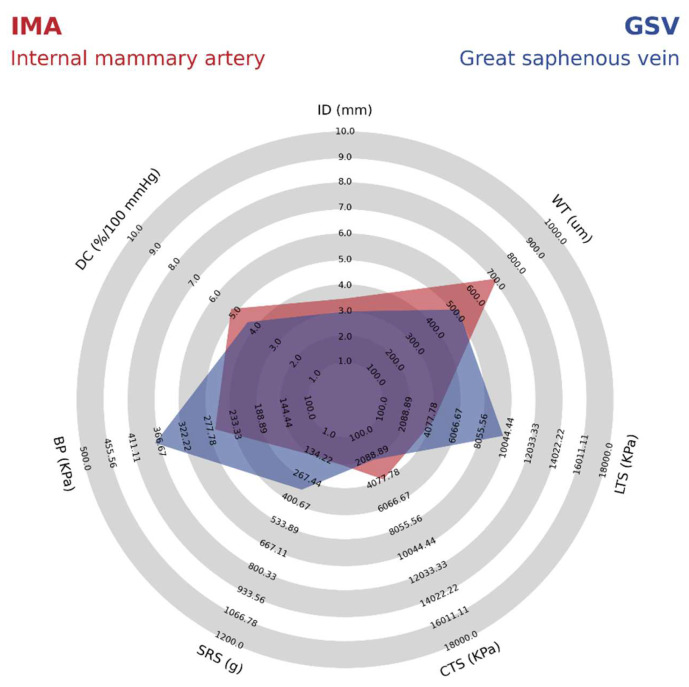
Comparison of the mechanical properties of the internal mammary artery (IMA) and great saphenous vein (GSV) as vascular grafts. Identified mechanical properties include internal diameter (ID), wall thickness (WT), longitudinal tensile strength (LTS), circumferential tensile strength (CTS), suture retention strength (SRS), burst pressure (BP), and dynamic compliance (DC). Chart generated from the Python library developed by StatsBomb/Anmol Durgapal (accessed on 29 May 2022) [29].

**Figure 4 polymers-14-03440-f004:**
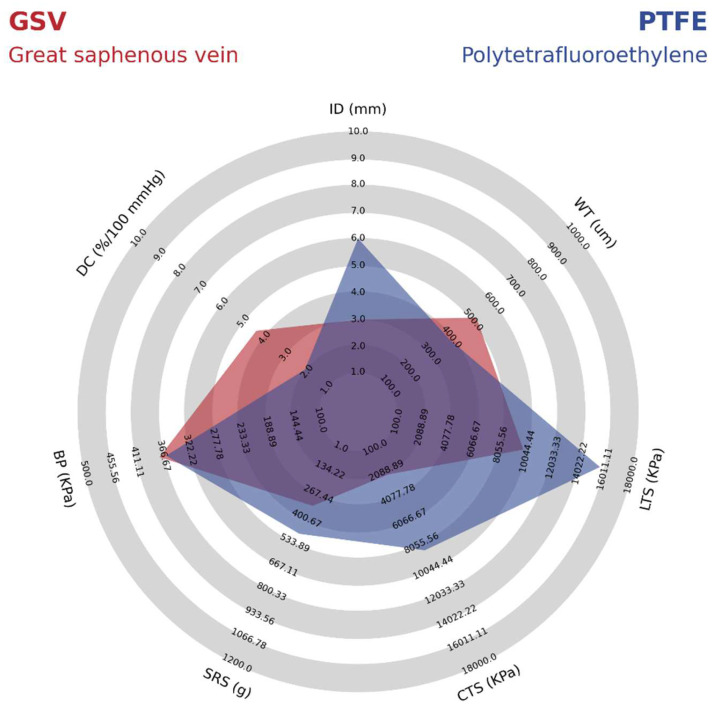
Mechanical properties comparison of the great saphenous vein, as a gold standard, and PTFE. Mechanical properties identified include Internal diameter (ID), wall thickness (WT), longitudinal tensile strength (LTS), circumferential tensile strength (CTS), suture retention strength (SRS), burst pressure (BP), and dynamic compliance (DC). Chart generated from the Python library developed by StatsBomb/Anmol Durgapal (accessed on 29 May 2022) [29].

**Figure 5 polymers-14-03440-f005:**
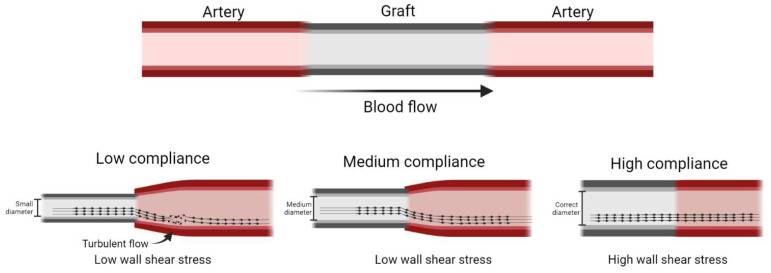
Effect of compliance on flow patterns and wall shear stress at the distal anastomosis in a VGs. Data retrieved from Post A, et al. [45] Image created with BioRender.com (accessed on 29 May 2022).

**Figure 6 polymers-14-03440-f006:**
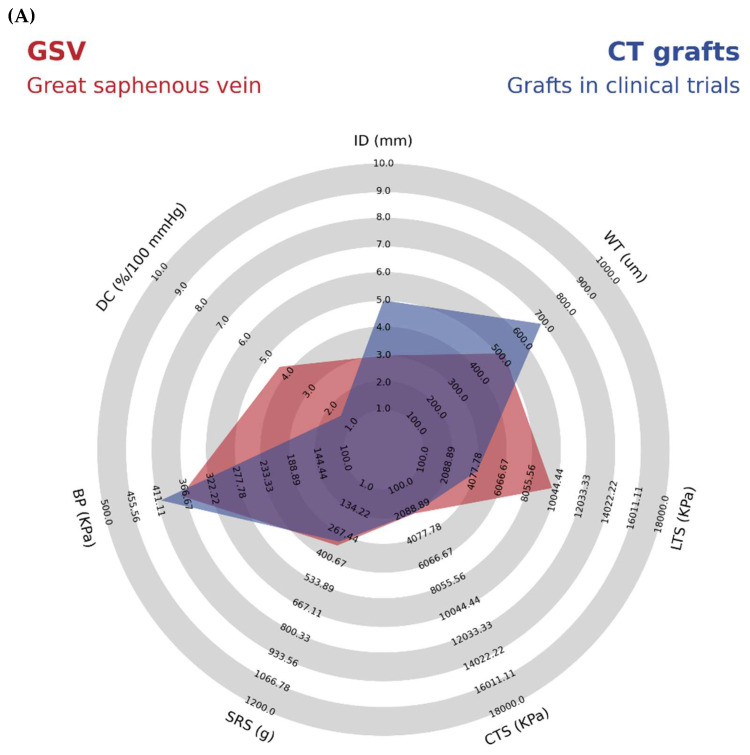
(**A**) Mechanical properties comparison of great saphenous vein, as a gold standard, and grafts in clinical trials. Mechanical properties identified include internal diameter (ID), wall thickness (WT), longitudinal tensile strength (LTS), circumferential tensile strength (CTS), suture retention strength (SRS), burst pressure (BP), and dynamic compliance (DC). Chart generated from the Python library developed by StatsBomb/Anmol Durgapal (accessed on 29 May 2022) [29]. (**B**) Mechanical properties comparison of great saphenous vein, as a gold standard, internal mammary artery, and each graft made in clinical trials. Circumferential tensile strength (CTS), longitudinal tensile strength (LTS), suture retention strength (SRS), internal mammary artery (IMA), and great saphenous vein (GSV). Data are according to literature reports of a representative data set. Additional information regarding descriptive analysis can be found in Appendix A. Graph generated with GraphPad Prism (accessed on 29 May 2022).

**Figure 7 polymers-14-03440-f007:**
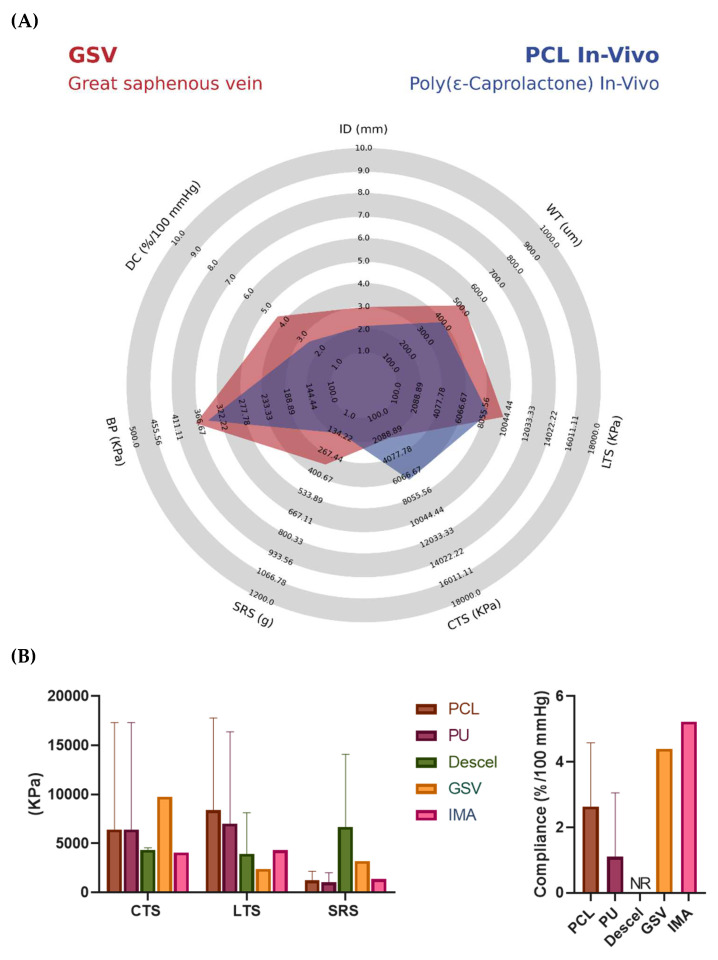
(**A**) Mechanical properties comparison of great saphenous vein, as a gold standard, and Poly(ɛ-Caprolactone) tested in vivo. Mechanical properties identified include internal diameter (ID), wall thickness (WT), longitudinal tensile strength (LTS), circumferential tensile strength (CTS), suture retention strength (SRS), burst pressure (BP), and dynamic compliance (DC). Chart generated from the Python library developed by StatsBomb/Anmol Durgapal (accessed on 29 May 2022). (**B**) Mechanical properties comparison of great saphenous vein, as a gold standard, internal mammary artery and each graft made in clinical trials. Circumferential tensile strength (CTS), longitudinal tensile strength (LTS), suture retention strength (SRS), Decellularized (Descel), internal mammary artery (IMA), and great saphenous vein (GSV). Data are shown as mean ± standard deviation, according to the number of studies reporting the data. Additional information regarding descriptive analysis can be found on Appendix A. Graph generated with GraphPad Prism (accessed on 29 May 2022).

**Figure 8 polymers-14-03440-f008:**
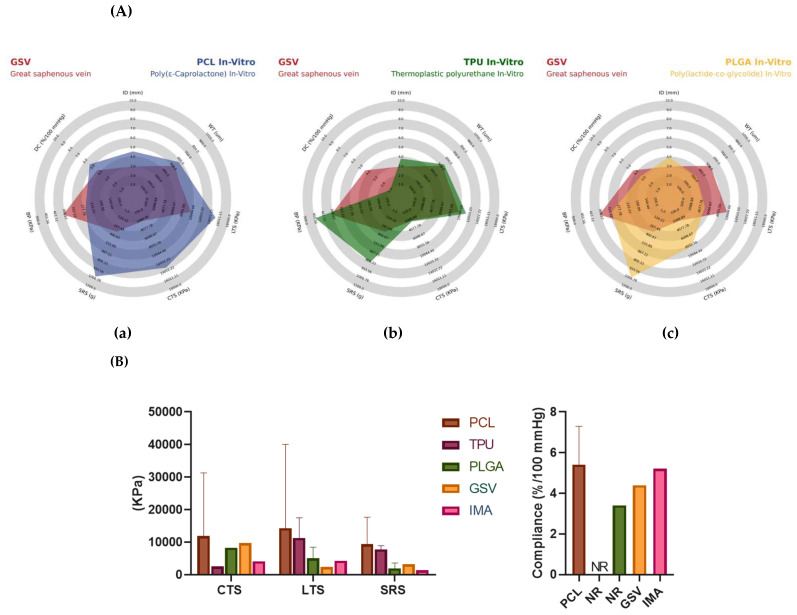
(**A**) Mechanical properties comparison of great saphenous vein, as a gold standard, (**a**) PCL, (**b**) TPU, and (**c**) PLGA tested in vitro. Mechanical properties identified include internal diameter (ID), wall thickness (WT), longitudinal tensile strength (LTS), circumferential tensile strength (CTS), suture retention strength (SRS), burst pressure (BP), and dynamic compliance (DC). Chart generated from the Python library developed by StatsBomb/Anmol Durgapal (accessed on 29 May 2022). (**B**) Mechanical properties comparison of great saphenous vein, as a gold standard, internal mammary artery and PCL, TPU, and PLGA subgroups of in vitro studies. Circumferential tensile strength (CTS), longitudinal tensile strength (LTS), suture retention strength (SRS), internal mammary artery (IMA), and great saphenous vein (GSV). Data are shown as mean ± standard deviation, according to the number of studies reporting the data. Additional information regarding descriptive analysis can be found on Appendix A. Graph generated with GraphPad Prism (accessed on 29 May 2022).

**Table 1 polymers-14-03440-t001:** Mechanical properties in blood vessels: internal mammary artery (IMA) and great saphenous vein (GSV).

Test Performed	IMA	GSV	Reference
Internal diameter (mm)	3.50	3	[22,23]
Wall Thickness (µM)	710	518	[24,25]
Circumferential tensile Strength (KPa)	4100	2405	[12,19,26]
Longitudinal tensile Strength (KPa)	4300	9760	[12,19,26]
Burst pressure (KPa)	266	371.96	[12,26]
Suture Retention Strength (g)	138	327	[22,27]
Dynamic Compliance (%/100 mmHg)	5.22	4.40	[12,28]

**Table 2 polymers-14-03440-t002:** Mechanical properties in average commercial PTFE vascular grafts for Arteriovenous Fistula with 6 mm internal diameter.

Test Performed	Average Value	Reference
Relaxed internal diameter (mm)	6 ^2^	[36]
Pressurized Internal Diameter (mm)	6 ^2^	[36]
Wall Thickness (mm)	0.39 ^2^	[36]
Porosity (%)	50	[37]
Void area (µM) ^1^	20–500	[37]
Water permeability (mL·cm^2^/min)	4320	[37]
Circumferential tensile Strength (KPa)	16,000 ^2^–20,590 ^3^	[37]
Longitudinal tensile Strength (KPa)	15,630 ^2^–41,480 ^3^	[37,38]
Burst Strength (kPa)	361	[37,38]
Suture Retention Strength (g)	480	[37]
Dynamic Compliance (%/mmHg)	2.1	[37]
Strength After Puncture (KPa)/ # Punctures	19,220/0 9230/8 8730/16 6370/24	[27]

^1^ Minimum value and maximum value to maintain tissue integration. ^2^ Non-stretchable PTFE VGs. ^3^ Stretchable PTFE VGs.

**Table 3 polymers-14-03440-t003:** Effect pf baseline and low WSS over endothelial cells, smooth muscle cells, and macrophages gene expression.

		Endothelial Cells Genes		Smooth Muscle Cells Genes		Macrophage Genes	
Wall Shear Stress (N/m^2^)		Down-Regulated	Up-Regulated	Down-Regulated	Up-Regulated	Down-Regulated	Up-Regulated
Physiological Limits	1.5–2.4	E-Selectin	KLF-2	P21	Ciclyn D1	Leukocyte	M2 phenotype
		TXNIP	ERK	BMP4	AKT		CD206
		PKC	P38	SMAD	IL-11		IL-10
		JNK	ERK				TGF-β1
		TNFa	CD31				
			eNOS				
			vWF				
Low limits	0.1–1	eNOS	VCAM-1	a-SMA	Prolifferative	M2 phenotype	M1 Phenotype
		NOX4	ICAM-1	SM22	MMP2		NF-κB
		NOTCH1	EDN-1	SMTN	TGF-β1		IL-1
			MCP-1	CNN	PDGF		MCP-1
			PDGF				Selectin
							MMP2-9

Data obtained and complemented from Rodriguez-Soto, et al. [14].

**Table 5 polymers-14-03440-t005:** Mechanical properties in blood vessels: Poly(ɛ-Caprolactone) tested in vivo (PCL in vivo) and great saphenous vein (GSV).

Test Performed	PCL In Vivo	GSV	Reference
Internal Diameter (mm)	2.17	3	[22,23,67,68,70,72,74,76,78,79,80,81,82,83,88,89,90,95,96,97,98,99,102,103,104,105]
Wall Thickness (µM)	400	518	[24,25,67,68,70,72,74,76,78,79,80,81,82,83,88,89,90,95,96,97,98,99,102,103,104,105]
Circumferential Tensile Strength (KPa)	6440	2405	[12,19,26,67,70,74,76,78,88,89,90,92,96,98]
Longitudinal Tensile Strength (KPa)	8400	9760	[11,19,26,67,68,72,74,76,78,79,81,82,83,95,97,102,103,104]
Burst Pressure (KPa)	348.22	371.96	[12,26,67,74,76,79,81,89,90,97,99,102,104,105,107]
Suture Retention Strength (g)	124	327	[22,27,67,74,76,95,96,99,104,107]
Dynamic Compliance (%/100 mmHg)	2.63	4.40	[12,28,74,76,78,81,104]

**Table 6 polymers-14-03440-t006:** Mechanical properties in blood vessels: PCL, TPU, and PLGA tested in vitro (PCL in vivo) and great saphenous vein (GSV).

Test Performed	PCL In Vitro	TPU In Vitro	PLGA In Vitro	GSV	Reference
Internal Diameter (mm)	4.59	3.83	4	3	[22,23,113,115,117,120,122,123,124,125,132,133,139,140,141,142,145,154,157]
Wall Thickness (µM)	580	540	330	518	[24,25,113,115,117,118,120,123,124,133,140,141,143,145,147,151]
Circumferential Tensile Strength (KPa)	13,360	2570	8350	2405	[12,19,26,113,115,120,122,125,141,143,145,151]
Longitudinal Tensile Strength (KPa)	15,170	11,320	5010	9760	[12,26,115,117,118,122,124,127,132,135,137,139,140,141,142,143,145,147,151,154]
Burst Pressure (KPa)	264.51	451.29	312.84	371.96	[12,26,118,120,122,123,142,143,147,151,154,157]
Suture Retention Strength (g)	1034	803	1050	327	[22,27,115,117,118,123,143,151]
Dynamic Compliance (%/100 mmHg)	5.42	NR	3.41	4.40	[12,28,117,120,124]

## Data Availability

Not applicable.

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
