# Peer review of "Small Diameter Cell-Free Tissue-Engineered Vascular Grafts: Biomaterials and Manufacture Techniques to Reach Suitable Mechanical Properties"

_polymers, 2022, doi:10.3390/polym14173440_

Round 1
Reviewer 1 Report
1. Include more references for Sections 4.1, 7.1 and 7.2 (Line 611)
2. The article goes into detail regarding how and why PTFE is widely employed in the fabrication of small diameter vascular grafts, with the argument that since it is non-biodegradable, other alternate materials that are bio-degradable should be explored. However, there is not much detail on what bio-degradable alternatives are preferred and why. Please incorporate a section to discuss most commonly used biodegradable alternatives (as stated in the Supplementary material) including PCL and PU for the fabrication of TEVGs including their advantages and pitfalls as reported in literature.
3. Have a separate section captioned as "Approaches/Techniques for fabrication of small diameter TEVGs" and include the content from Line 638 to Line 668 under that. Discuss other methods that are widely used today such as 3D bioprinting and Decellularization including Electrospinning, Freezedrying and Solvent casting which are already stated. Use the following articles as references:
* Durán-Rey, David, Verónica Crisóstomo, Juan A. Sánchez-Margallo, and Francisco M. Sánchez-Margallo. "Systematic Review of Tissue-Engineered Vascular Grafts." Frontiers in bioengineering and biotechnology 9 (2021).
* Carrabba M, Madeddu P. Current Strategies for the Manufacture of Small Size Tissue Engineering Vascular Grafts. Front Bioeng Biotechnol. 2018 Apr 17;6:41. doi: 10.3389/fbioe.2018.00041. PMID: 29721495; PMCID: PMC5916236.
* Wang, Ziyu, Suzanne M. Mithieux, and Anthony S. Weiss. "Fabrication techniques for vascular and vascularized tissue engineering." Advanced Healthcare Materials 8, no. 19 (2019): 1900742.
* Weekes, Angus, Nicole Bartnikowski, Nigel Pinto, Jason Jenkins, Christoph Meinert, and Travis J. Klein. "Biofabrication of small diameter tissue-engineered vascular grafts." Acta biomaterialia (2021).
* Jafarihaghighi, Farid, Mehdi Ardjmand, Abolfazl Mirzadeh, Mohammad Salar Hassani, and Shahriar Salemi Parizi. "Current challenges and future trends in manufacturing small diameter artificial vascular grafts in bioreactors." Cell and Tissue Banking 21, no. 3 (2020): 377-403.
4. Please do a thorough grammar check using Grammarly or a similar tool.
Author Response
Dear Dr.
Thank you so much for your revisions and suggestions, we hope that the new version fits your expectations.
Regards,
María

Reviewer 2 Report
The problem considered at work is very important and undertaken for many years by scientists. The article deals with the growing need for vascular grafts. The problem is current and very significant in light of the continuous increase in the new materials and production methods.
The manuscript is interesting. However, there are several points that I would like to address:
1. In the manuscript, there is no explanation of what exactly means "suitable mechanical properties". It would be interesting to include clear/precise parameters, and properties of such kinds of materials, which are taken into account during the manufacturing process.
2. The title of the second chapter can be changed to be more adequate to the subject matter it covers, i.e. the selection and review process of available databases.
3. Figure 2- there is no explanation of the abbreviation “FBs” used, neither in the title of the figure nor in the text of the manuscript.
4. From a biochemical point of view, a very interesting problem is the influence of surface properties (roughness, porosity) on bioadhesion problems, i.e. forming biological adhesive layers and then biofilm. This is very important for possible infections. It would be interesting to summarize what is already known on this topic, taking into account, both biomaterials and manufacturing methods.
5. There are some stylistic and editing errors and the manner of writing, e.g. H2O2 - numbers should be in the form of subscripts.
Author Response
Dear Dr.
Thank you so much for your kind comments and revisions, we have modified the manuscript according to your suggestions and we hope that it fits yours expectations.
Hope to hear from you,
María

Round 2
Reviewer 1 Report
Good job with incorporating most of the reviewer comments. I have one final suggestion: please move the two paragraphs on Electrospinning under Section 7.2 to 7.5 as Section 7.2 talks about the materials of choice while 7.5 talks about the approaches, this would be more appropriate.
Author Response
Dear Dr.
Thank you for your suggestion, we are glad that you find our corrections propper.
Please find the document with the suggested changes.

Reviewer 2 Report
All my comments were improved in the revised version of the manuscript. I endorse it for publication.
Author Response
Dear Dr.
Thank you so much, we are glad that you have found the issues addressed, and that you have liked the manuscript